# Perinatal Lead (Pb) Exposure and Cortical Neuron-Specific DNA Methylation in Male Mice

**DOI:** 10.3390/genes10040274

**Published:** 2019-04-04

**Authors:** John F. Dou, Zishaan Farooqui, Christopher D. Faulk, Amanda K. Barks, Tamara Jones, Dana C. Dolinoy, Kelly M. Bakulski

**Affiliations:** 1Department of Epidemiology, School of Public Health, University of Michigan, Ann Arbor, MI 48109, USA; johndou@umich.edu; 2Department of Pediatrics, Cincinnati Children’s Hospital Medical Center, Cincinnati, OH 45229, USA; zafa@umich.edu; 3Department of Environmental Health Sciences, School of Public Health, University of Michigan, Ann Arbor, MI 48109, USA; trmeier@umich.edu; 4Department of Animal Science, College of Food, Agricultural, and Natural Resource Sciences, University of Minnesota, St. Paul, MN 55108, USA; cfaulk@umn.edu; 5Department of Pediatrics, University of Minnesota Masonic Children’s Hospital, Minneapolis, MN 55454, USA; barks012@umn.edu; 6Department of Nutritional Sciences, School of Public Health, University of Michigan, Ann Arbor, MI 48109, USA

**Keywords:** lead, Pb, DNA methylation, neuron, in utero

## Abstract

Lead (Pb) exposure is associated with a wide range of neurological deficits. Environmental exposures may impact epigenetic changes, such as DNA methylation, and can affect neurodevelopmental outcomes over the life-course. Mating mice were obtained from a genetically invariant C57BL/6J background agouti viable yellow *A^vy^* strain. Virgin dams (*a*/*a*) were randomly assigned 0 ppm (control), 2.1 ppm (low), or 32 ppm (high) Pb-acetate water two weeks prior to mating with male mice (*A^vy^*/*a*), and this continued through weaning. At age 10 months, cortex neuronal nuclei were separated with NeuN^+^ antibodies in male mice to investigate neuron-specific genome-wide promoter DNA methylation using the Roche NimbleGen Mouse 3x720K CpG Island Promoter Array in nine pooled samples (three per dose). Several probes reached *p*-value < 10^−5^, all of which were hypomethylated: 12 for high Pb (minimum false discovery rate (FDR) = 0.16, largest intensity ratio difference = −2.1) and 7 for low Pb (minimum FDR = 0.56, largest intensity ratio difference = −2.2). Consistent with previous results in bulk tissue, we observed a weak association between early-life exposure to Pb and DNA hypomethylation, with some affected genes related to neurodevelopment or cognitive function. Although these analyses were limited to males, data indicate that non-dividing cells such as neurons can be carriers of long-term epigenetic changes induced in development.

## 1. Introduction

Exposure to lead (Pb) is associated with a wide range of neurological deficits. Early life Pb exposure is associated with neurobehavioral impairment in children [1,2], who exhibit worse global executive function, metacognition, and behavioral regulation as Pb levels rise. The US Centers for Disease Control and Prevention estimate that approximately 500,000 children age 1–5 years currently have blood Pb levels above the reference value (≥5 μg/dL) [3]; however, no safe level of Pb has been identified. In adults, Pb exposure is linked to cognitive decline [4], amyloid lateral sclerosis [5], and Parkinson’s disease [6]. Exposure to Pb exerts adverse effects on learning and memory ability in young rats [7,8], mice [9,10], and monkeys [11,12]. Environmental perturbations early in life can affect pathophysiology in adulthood [13,14,15,16,17,18], as described in the Developmental Origins of Health and Disease (DOHaD) paradigm. When a given exposure ceases, its effects may persist, as seen in studies of children whose mothers were exposed to famine conditions while in the first trimester of pregnancy [19].

Although the mechanisms by which these alterations are programmed are largely unknown, DNA methylation has emerged as a major candidate for investigation. For example, maternal licking and grooming behavior in rats affects offspring glucocorticoid receptor expression via differential hippocampal DNA methylation of the gene *Nr3c1*, altering offspring stress response [20]. Our group observed female mice exposed to bisphenol A (BPA) in utero display dose-dependent differences in DNA methylation in tail tissue at three weeks of age [21] and significantly increased hyperactivity later in life [22]. In mice perinatally exposed to Pb, several intracisternal A particle retrotransposons have altered DNA methylation in brain and kidney with tissue and sex dependent effects [23]. Pb acts indirectly to produce reactive oxygen species [24], which alter the binding of transcription factors like Sp1, and other regulatory factors that are mediated by DNA methylation, such as methyl-CpG binding protein 2 (MeCP2) [25]. Environmentally sensitive stable epigenetic changes, such as DNA methylation, may affect neurodevelopmental outcomes over the life-course.

In post-mitotic cells, DNA methylation patterns are often highly stable epigenetic markers that govern expression and tissue specificity [26,27]. Neuronal cells are peculiar in that they have approximately the same life span as the individual and they make and break synaptic connections well into adult life [28,29]. Neuronal activity modulates the DNA methylation landscape [30,31], and reciprocally, DNA methylation modulates synaptic transmission and excitability of neurons [32,33]. Given their distinct function, isolated neuronal cells have a markedly distinct methylation profile from bulk cortical tissue [34]. Studies of disease or exposure in bulk cortical tissue have observed small magnitude differences in DNA methylation [35]. Thus, it is highly likely that neurons, which make up approximately 65% of the mouse brain cell population [36], may accumulate changes in methylation in response to environmental exposures that are distinct or in greater magnitude from changes seen in bulk cortical tissue.

Neuronal cells may be vehicles through which early-life Pb exposure exert lasting effects. Persistent epigenetic changes could be carried through the long lifespans of neurons. However, no studies specifically investigate the association between neuron-specific genome-wide DNA methylation and early-life Pb exposure. Here, we have isolated the mouse cortex from adult male mice developmentally exposed to leaded or control water and used an established protocol to separate NeuN^+^ (a specific marker for neuronal nuclei) neuronal cells from NeuN^−^ non-neuronal cells in mice exposed to Pb perinatally. DNA methylation patterns were assessed at 10 months of age in pooled samples (N = 3 pools per exposure group).

## 2. Materials and Methods

### 2.1. Mouse Study Population

Mating mice were obtained from a C57BL/6J background agouti viable yellow *A^vy^* strain [37] maintained for over 220 generations with forced heterozygosity for the *A^vy^* allele through the male line, resulting in a genetically invariant background with 93% similarity to C57BL/6J [38,39]. Virgin dams *(a*/*a)* were assigned randomly to one of three treatment groups with exposure to Pb-acetate through the drinking water: None, 2.1 ppm (low) and 32 ppm (high), corresponding to peak maternal blood levels of approximately 2.5 and 25 μg/dL, respectively [39]. Offspring blood lead levels were not measured to avoid an additional stress exposure. Lead exposure began two weeks prior to mating with viable yellow agouti male mice (*A^vy^*/*a*) and continued throughout gestation to three weeks after birth, at which time Pb-exposed mice were weaned onto Pb-free control water. Breeding was designed to produce 50% *A^vy^*/*a* offspring and 50% *a*/*a* offspring. The ectopic expression of the *agouti* gene in *A^vy^* mice is known to affect adult onset obesity and tumorigenesis [40], which may confound the relationship between Pb and DNA methylation. Therefore, we conducted the mouse cortex epigenetic analysis on *a*/*a* offspring only. This study sample was restricted to only males. The developmentally exposed Pb and control offspring used in this study were sacrificed at 10 months of age to extract the cortex from whole brain.

### 2.2. Sample Ascertainment and Preparation

The cerebral cortex was dissected from the brain on dry ice. Approximately 200 mg of cortical tissue was obtained from each mouse, consistent with previously described methods for neuronal nuclei separation [34,41]. To prepare for fluorescence-activated cell separation (FACS), we first minced cortical tissue using razor blades. To detach cells, each 200 mg portion was incubated with 1 mL Accutase (Sigma-Aldrich, St. Louis, MO, USA) for 30 min, then centrifuged briefly. Next, we replaced the supernatant with 1.0 mL of Hibernate A medium (Thermo Fisher Scientific, Waltham, MA, USA). This was subsequently triturated through a ~0.8 mm glass pipette once, and a ~1.0 mm glass pipette twice. This mixture was strained through a 100-micron strainer, then through a 40-micron strainer (BD, Franklin Lakes, NJ, USA). To separate cells from extracellular matrix and other connective tissue, the mixture was layered on a discontinuous density gradient of Percoll (Sigma-Aldrich) with 1 mL layers that were 12%, 18%, and 24% Percoll, diluted in Hibernate A and 22 mM NaCl. We then centrifuged at 500 rcf for 10 min. The bottom and middle layers were then fixed and permeabilized by incubation in a 1:1 ratio of ice-cold 100% ethanol for 20 min at 4 °C. Afterwards, we replaced the EtOH solution with Hibernate A.

Following cell fixation, permeabilization and nuclear fractionation, the nuclei were now separated and ready for immunolabeling. We used a monoclonal mouse anti-NeuN (clone A60, from Millipore) antibody that was pre-conjugated with Alexa Fluor 488. All samples were blocked in 1 mL of a 1% BSA/10% goat serum solution for one hour. The main sample to be sorted via FACS was incubated in a 1:20,000 solution of Anti-NeuN 488 antibody in the dark for 30 min, then washed in PBS solution five times at approximately 450 rcf for five minutes per wash to remove unbound antibody. Two control samples were implemented to ensure proper interpretation of FACS scatter diagrams. Approximately 100 μL of the nuclei solution was unlabeled to determine baseline scatter profile of cell relative to remaining debris. An additional 20 μL was used as a saturated binding control. Non-fluorescent anti-NeuN was added in a 10:1 ratio prior to addition of fluorescent anti-NeuN to saturate specific binding sites. Under the assumption that specific binding has greater fluorescence, we used the fluorescence from non-specific binding control to set the gate for neurons specifically bound to anti-NeuN. The antibody we employed was pre-conjugated, so there was no need for a secondary binding control. FACS sorting of nuclei was performed on either FACS Aria or MoFlo Astrios, with assistance from the Flow Cytometry Cores at the University of Michigan Biomedical Research Science Building and Cancer Center.

DNA was extracted from cells by modifying previously established methods for DNA extraction from blood cells, described in Qiagen (Hilden, Germany) protocols. All reactions were scaled appropriately, but are reported here for 1 mL of FACS-sorted NeuN^+^ sample. Each sample was incubated with 100 μL of Proteinase K and 1.2 mL of proprietary Qiagen lysis buffer at 50 °C overnight. One equivalent of 100% frozen ethanol was added, followed by centrifugation through a Qiagen DNA binding column. The solution was then washed three times with Qiagen wash buffer, and eluted in ~200 μL of 10 mM Tris/0.5 mM EDTA buffer. We used neuronal NeuN^+^ genomic DNA for tiling array hybridization. Sample pooling was necessary to reach the amount of genomic DNA necessary to perform hybridization, approximately 9 μg. The pooling design is shown in Appendix A. Briefly, three pooled samples were used for each control, low, and high Pb exposure group, resulting in nine total pooled samples. Each pool consisted of DNA from at least two isogenic mice.

### 2.3. Nimblegen Tiling Array Sample Preparation

Genomic DNA from pooled NeuN^+^ DNA samples was sonicated to fragment sizes ranging from 200–1000 bp using an Episonic 1100-series sonicator (Farmingdale, NY, USA). DNA (7.5–9 μg) was sonicated in cycles of 15 s-ON and 30 s-OFF for 15 min in 8–20 °C water. For each pool, one sonicated sample was enriched for methylated DNA, while another sample was used as a control genomic input for co-hybridization. We enriched fragmented samples for methylated DNA through methyl-CpG binding domain-based capture using the EpiMark Methylated DNA Enrichment Kit (New England Biolabs, Ipswich, MA, USA). To obtain sufficient amounts of DNA for hybridization, 10 ng of the captured DNA was subject to whole genome amplification using the GenomePlex Complete Whole Genome Amplification Kit (Sigma-Aldrich, St. Louis, MO, USA).

The un-enriched sample and CpG methylation-enriched sample—henceforth referred to as the “experimental” sample—were labeled with Cy3 and Cy5 dye, respectively, using the NimbleGen Dual-Color Labeling Kit (Roche NimbleGen, Madison, WI, USA) following the protocol outlined in the NimbleGen Array User Guide (NimbleGen Array User Guide *DNA Methylation Arrays*, Version 7.2). These labeled fractions were pooled together in equivalent amounts and co-hybridized to the Roche NimbleGen Mouse DNA methylation 3x720K CpG island Promoter Array for 16–20 h. Array slides contain three subarrays, each of which holds 720,000 probes scanning 15,980 CpG islands in 20,404 murine gene promoters. After hybridization, arrays were washed and scanned using a 2 μm-resolution scanner (NimbleGen MS 200 Microarray Scanner, used courtesy of Dr. Thomas Glover, Department of Human Genetics).

### 2.4. Bioinformatics Processing

Scanned images were uploaded to Nimblegen DEVA Software, version 1.2.1, to extract location and raw Cy3 and Cy5 intensity—corresponding to control and experimental sample, respectively—of each feature. Locally weighted polynomial regression (LOESS) spatial correction was performed to correct for position-dependent non-uniformity of signals within the sub-array. After position-dependent normalization, the DEVA software calculated log_2_ (Cy5-labeled experimental/Cy3-labeled control) ratios for each probe. Probes were centered around Tukey’s biweight mean for all probes. Probes included in analysis were restricted to non-control probes mapped to chromosomes, leaving 707,998 probes. In DEVA, probes were annotated to genes overlapping within a range of 1000 downstream to 5000 upstream base pairs. 

### 2.5. Probe-Level and Pathway Analysis

R Statistical Software (version 3.5.1) was used to fit basic linear regression models for probe-associated log_2_(ratios), with Pb exposure categorically modeled using the no exposure group as reference. We applied empirical Bayes smoothing to the standard errors. Genes were ranked based on *p-*values from the linear regression model. Using the R package fgsea [42], we performed gene set enrichment analysis for gene ontology biological pathways with 1000 permutations.

### 2.6. Regional Analysis

We used the statistical package DMRcate [43] to test associations between Pb exposure and gene promoter methylation at the regional level. We ranked regions based on the Stouffer transformed *p*-value of the group of false discovery rates FDRs for differentially methylated region constituent CpGs. The Gaussian kernel bandwidth lambda value used was the default of 1000, and minimum consecutive probes in a differentially methylated region was set to four.

### 2.7. Data Availability

Raw and processed data generated in this study can be accessed from the Gene Expression Omnibus (accession number GSE125174).

## 3. Results

### 3.1. Neuronal Separation

Neuronal nuclei were separated for 19 cortex samples (six each from control and low Pb exposure, and seven from high Pb exposure resulting in three biological replicate pools per exposure group; see Appendix A for pooling scheme). Debris was removed based on the profile of forward scatter-area/side scatter-area (FSC-Area vs. SSC-Area). Labeled nuclei that were not residual debris comprised 63.12% of the initial sample (Appendix A, Gate 1). Doublets were removed by first using scatter profile of forward scatter-width vs. forward scatter-height (FSC-Width vs. FSC-Height), then by side scatter-width vs. side scatter height (SSC-Width vs. SSC-Height) (Appendix A). Fractionation gates for NeuN^+^ and NeuN^−^ sample were set using profiles of 525/50 488-Area fluorescence vs. SSC-Area. There was an auto-fluorescence rate of 0.08% for a sample of 10,000 cells (Appendix A, Gate R4). The R5 gate was further adjusted in the sample that was pre-labeled with non-fluorescent anti-NeuN antibody (Appendix A). For a representative cortex sample, we observed 74.36% of nuclei were NeuN^+^, and 22.83% of nuclei were NeuN^−^ (Appendix A). These proportions were similar for all samples sorted. The average number of nuclei per sorted sample was 1.21 × 10^6^ NeuN+ nuclei and 4.65 × 10^5^ NeuN^−^ nuclei. The average amount of DNA extracted was 4.72 μg per one million nuclei. Three pooled NeuN^+^ DNA samples were sonicated to fragment the DNA (Appendix A). Pooled samples 3 and 7 were re-sonicated to achieve appropriate fragment size range (Appendix A). Next, samples were measured for genome-wide DNA methylation from each treatment group. 

### 3.2. Probe-Level Analysis

We tested for individual probe level differences in DNA methylation between Pb exposure and control separately for the low and high Pb groups. High Pb exposed mice had 12 differentially methylated probes reach *p*-value < 10^−5^ (minimum FDR value = 0.16, largest intensity ratio difference = −2.1). All 12 of these probes were hypomethylated in lead exposure compared to controls. At a lesser threshold of *p*-value < 0.01, there were 20,774 probes differentially methylated relative the control group (largest intensity ratio difference = −2.7), with 64.9% hypomethylated compared to controls. Low Pb exposed mice had seven differentially methylated probes reach *p*-value < 10^−5^ (minimum FDR value = 0.56, largest intensity ratio difference = −2.2). All seven probes were also hypomethylated compared to controls. At the *p*-value < 0.01 threshold, there were 4662 probes differentially methylated relative to controls (largest intensity ratio difference = −2.3). Of these probes, 53.6% had negative intensity ratio difference, indicating hypomethylation with exposure relative to controls. Differentially methylation testing for the high or low Pb groups related to controls had 1087 probes (*p-*value < 0.01) in common. Full probe results for high and low Pb exposure are in Appendix A, respectively.

### 3.3. Regional Analysis

Probe-level data were combined with genomic location information to test for differentially methylated regions by each Pb exposure group. No regions met genome-wide testing criteria for differential DNA methylation by Pb exposure. We observed six regions that were associated with high Pb exposure relative to controls that met a Stouffer cutoff of 0.5 and had the lowest *p*-values (Figure 1). The region that was most highly differentially methylated was most closely linked with predicted pseudogene 7609 (*Gm7609*). The regions associated with *Gm7609,* RIKEN cDNA D130017N08 gene (*D130017N08Rik*), artemin (*Artn*), zinc finger protein 974 (*Zfp974*), predicted gene 2012 (*Gm2012*), and complement component 5a receptor 1 (*C5ar1*) were all hypomethylated on average across the region in the mice exposed to 32 ppm Pb in drinking water relative to the control. When considering all regions returned by DMRcate (*n* = 20,564) (Appendix A), 58% of regions showed mean hypomethylation for high Pb exposure compared to control. 

When comparing the low Pb dose group relative to controls, no regions had Stouffer *p*-value < 0.988. Several regions had Stouffer values of approximately 0.9881, and the six regions with the largest mean ratio difference are shown in Figure 2. Of these regions, the one most differentially methylated was linked to G protein-coupled receptor 20 (*Gpr20*). Five regions linked to *Gpr20*, predicted gene 10471 (*Gm10471*), zinc finger protein 787 (*Zfp787*), X-linked Kx blood group related 7 (*Xkr7*), and cadherin-like 24 (*Cdh24*) had mean hypomethylation, while the region linked to ankyrin repeat and death domain containing 1B (*Ankdd1b*) had mean hypermethylation. Out of all regions reported by DMRcate (*n* = 9549) (Appendix A), 55% showed mean hypomethylation over the region for low Pb exposure. 

Of the top 100 regions each for high Pb relative to control and low Pb relative to control, only three overlapped. The nearest genes to these three regions were paired-Ig-like receptor A1 (*Pira1*), single-strand selective monofunctional uracil DNA glycosylase (*Smug1*), and free fatty acid receptor 2 (*Ffar2*). In the region mapped to *Smug1,* high Pb dose had a larger estimated difference (mean ratio difference = −0.38, Stouffer = 0.58) than low Pb dose (mean ratio difference = −0.16, Stouffer = 0.989). In the two other regions, high Pb showed mean hypomethylation across the regions, while low Pb had mean hypermethylation (Figure 3).

### 3.4. Pathway Analysis

Gene ontology analysis was conducted to test for biological pathway patterns among differentially methylated regions in the exposed mice. After multiple-comparison corrections, no pathway met genome-wide criteria. Several pathways had unadjusted *p*-value < 0.01 for high Pb (Table 1) and low Pb (Table 2) exposure. High Pb exposure-related DNA methylation regions were enriched for response to X-ray, positive regulation of hormone secretion, and neural tube formation pathways. High Pb exposure related DNA methylation regions were depleted for positive regulation of hair follicle development and positive regulation of cholesterol biosynthetic process pathways. Low Pb exposure related DNA methylation regions were enriched for female pregnancy, neuropeptide signaling pathway, cellular lipid metabolic process, and carbohydrate biosynthetic process pathways.

## 4. Discussion

This is the first cortex neuron-specific genome-wide methylation analysis of mice that were exposed to physiologically relevant levels of Pb during the perinatal period. We demonstrated clear separation of anti-NeuN^+^ cells from anti-NeuN^−^ cells. We isolated approximately 0.63–1.74 × 10^6^ NeuN^+^ nuclei and 0.27–1.26 × 10^6^ NeuN^−^ nuclei, yielding DNA on a scale of micrograms for each sample. In our sample, following cell lysis and nuclei fractionation, neurons contributed a large (approximately 74%) proportion of nuclei in FACS results. This proportion is within range of mouse neuronal cell proportions previously reported; however, nuclei from various cell types may have remained intact at different rates through our sample preparation and our data may not directly reflect cell proportions in vivo. We observed suggestive DNA methylation differences at genomic regions for low dose (2.1 ppm) Pb and high dose (32 ppm) Pb relative to the control treatment. The genome-wide DNA methylation data presented in this study suggest Pb exposure at low and high doses may have different effects on methylation in neuron-specific cortex cell populations. 

We tested for differentially methylated regions by analyzing consecutive probes and observed little overlap between the signatures for high and low Pb dose groups. For high Pb doses, one of the top regions was related to the *ARTN* gene encoding the artemin protein, which supports survival of sensory and sympathetic neurons, as well as midbrain dopaminergic neurons in culture and in vivo [44,45]. Another region was related to the *C5aR1* gene. In an Alzheimer’s disease mouse model, knock out of *C5aR1* prevented spatial memory cognitive deficits at 10 months of age, likely through reduction of inflammatory responses [46]. However, these regions did not meet genome-wide significance criteria in our study, and we interpret these suggestive results with caution. Despite not reaching genome-wide significance criteria, this data provides a useful guidepost to direct future orthogonal validation [47], such as bisulfite sequencing.

Regions associated with low Pb exposure were considerably less significant than those observed with high Pb exposure. Reduced associations in the low Pb group could explain minimal overlap between the low Pb associated regions and the high Pb associated regions. One of the top regions for low Pb exposure in our study was *Ankdd1b*. Independent genome-wide association studies have highlighted this gene for genetic overlap between migraine and major depressive disorder [48]. Another gene with a related differentially methylated region in our study is cadherin-like 24, part of a family of type II cadherins expressed in brain. This protein has temporal and spatial patterns of expression throughout mouse central nervous system development [49]. Related classical cadherins may have a role in synaptic differentiation [50]. 

In both high and low Pb exposures, the top regions included a zinc finger protein. The zinc finger is a binding motif important in processes like DNA replication and transcription, and Pb competes for the zinc binding site [51]. Disruptions by Pb impact DNA binding of zinc finger proteins, and expression of target genes [52]. For both exposure groups, no pathways met genome-wide criteria after adjusting for multiple comparisons. However, among pathways that reached a nominal *p*-value < 0.01, neural tube formation in high Pb exposure and neuropeptide signaling in low Pb exposure suggest neurological pathways may be differentially methylated by Pb exposure. 

Interestingly, we observed that the top probe and regional hits showed a pattern of hypomethylation associated with Pb exposure. All probes with *p*-value < 10^−5^ for high (12 probes) and low (seven probes) Pb exposure were hypomethylated. A previous study in human embryonic stem cells found that during neural differentiation (induced via treatment with derived neurotrophic factor and other supplements), acute exposure to Pb caused changes in DNA methylation, and approximately 80% of changed regions showed hypomethylation [53]. On the other hand, chronic exposure to Pb through the entire embryonic stem cell neural differentiation procedure resulted in DNA methylation changes that were more balanced (56% of differentially methylated regions hypermethylated) [53]. Early life Pb exposure in mice reduced cerebral levels of proteins involved in maintenance of DNA methylation such as DNA Methyltransferase 1 (DNMT1) and Methyl-CpG Binding Protein 2 [54]. Pb exposure in a zebrafish model also inhibited DNMT1 activity and reduced global CpG methylation [55]. Other studies showed patterns of hypermethylation. In neocortex tissue of mice, exposure to Pb in utero was associated with more overall methylation and repression of gene expression [56]. Another study found the majority (>90%) of differentially methylated CpG sites in the hippocampus of Pb exposed female mice were hypermethylated with less effect in males, and that effects of Pb were stronger in hippocampal tissue compared to the cortex [57]. In contrast, in hippocampal tissue of rats the sex linked Pb exposure observations were different, where 91% of differentially methylated genes were hypomethylated in females, but only 17% in males [58]. However, unlike the previous study conducted in hippocampal tissue, we observed a pattern of hypomethylation in cortex neuronal cells in a male-only sample. A limitation of our study is it did not include female mice, so we are unable to examine sex differences. The predominance of hypomethylation we observed mirrors some previous research in Pb exposure and brain DNA methylation, but future studies should further explore the potential sex-specific effects of Pb. 

Overall, we observe only weak associations between early Pb exposure and DNA hypomethylation. At best, these Pb associated methylation changes may be involved in putting mice at risk for developing later cognitive abnormalities. Pb impacts the brain through DNA methylation mechanism, as well as through interaction with calcium ion dependent processes and oxidative damage [59]. Pb exposure also affects neural phenotypes by changing brain structure, which may proceed or follow molecular mechanisms. In rats, perinatal Pb exposure induced pathological changes in nerve endings [60]. Cumulative Pb exposure may also be an important consideration. In human adults, cumulative occupational Pb exposure was associated with poorer performance in neuropsychological tests [61], and bone Pb concentration, reflecting cumulative exposure, is associated with faster declines in cognition test performance [4]. Pb exposure has clear brain impacts across the life course. In early life, Pb is associated with developmental delays, behavioral problems, and stunted growth, while in later life past Pb exposure is associated with cognitive decline [62].

While exposure to environmental toxicants may cause permanent damage to brain structures early in life, there is also ample animal model evidence that exposure to Pb can cause latent molecular changes that manifest much later. Rats [63] and monkeys [8] administered physiologically relevant levels of Pb early in life had an increase in an oxidative stress marker in the brain when measured later in the life. Oxidative stress, an imbalance in oxidant and antioxidant species within a system, is a significant mechanism in Alzheimer’s disease (AD) development [64]. Several genes in AD pathways are differentially expressed in monkeys exposed to Pb early in life [8]. *ADAM17* [65] and *APH1a* [66] are genes associated with amyloid precursor protein processing. These genes, and others, are implicated in AD [67,68,69,70,71]. Furthermore, in those aged monkeys a decrease in DNMT was also observed [8], suggesting possible epigenetic changes through methylation. Thus, it may be worthwhile to also examine DNA methylation in other time windows.

The present research is relatively rare in its analysis of a cell-type specific population within a brain region. Neurons are a particularly challenging set of cells in which to study DNA methylation. Neuronal activity modifies DNA methylation [30,31], suggesting that each individual cell may be differentially methylated depending on its activity. Thus, it is difficult to tell which regions of the methylome are activity-dependent, potentially directly modified by Pb, or prone to modification by Pb through its independent effect on neuronal activity. Future studies could differentiate between the activity-dependent neuronal methylome and the part of the methylome that may be less susceptible to neuronal firing. Additionally, consideration of neuronal subtypes may be important, as they have different methylation patterns [72,73]. If Pb targets specific subtypes, examination of the combined neuron population would attenuate observed effects. Neurons, as non-dividing cells, can be long-term carriers of epigenetic changes. Any DNA methylation patterns we observed would have persisted well into the life of the mouse at 10 months of age despite the exposure ending at weaning. For both high and low Pb exposures, males showed increased body weight at 10 months [39]. Effects on neurons may be potentiators of further environmental exposure due to behavioral control.

Our results are consistent with prior work published on the effect of early life exposure to Pb on the mouse DNA methylome [56]. The authors observed a correlation between expression and methylation, but they noted only a small percentage of genes were differentially methylated as a result of Pb exposure [56]. Some of the genes with altered expression in relation to Pb exposure in their study were related to some of the genes implicated in ours, namely SRY-box containing gene 4 and Ankyrin repeat domain 1. It is worth noting this previous study was performed in bulk tissue, not a specific population of neurons, and the mice were not exposed in utero [56]. Future directions of study could include correlation of DNA methylation with gene expression to expand the possible scope of biological inferences.

In the current study, our genome-wide analysis platform was a promoter tiling array. Although it provides widespread coverage of the genome, it focuses only on promoters. Other literature on the association between DNA methylation and various cancers clearly suggests that functional DNA methylation may be taking place in CpG shores, and not necessarily associated with promoter regions [74,75]. Additionally, in vitro Pb exposure on neuronal methylation affects regions beyond promoters [53]. Promoters and CpG islands are potentially less responsive to the environment and more tightly regulated. The NimbleGen platform used in the current study also demanded large quantities of DNA. The input genomic DNA required for genome-wide methylation studies is decreasing with advancing methylation-sequencing technologies. Future analyses may consider bisulfite sequencing with lower DNA input and greater genome coverage.

This study is the first to test a cortical neuron-specific population of cells of adult mice exposed to Pb in utero. The window of susceptibility to exposure was critical: all mice in our study were exposed to Pb via the maternal drinking water, and the exposure was cutoff at weaning. This unique exposure paradigm allowed us to test the influence of environmental exposures on the DNA methylome during neurodevelopment. We examined DNA methylation at 10 months of age, months after exposure ceased, and our observations likely represent stable changes in response to Pb exposure. The tiling array platform used in these studies was a discovery tool to identify pathways and regions that may be differentially methylated in utero dependent on environmental exposure. The present study has generated several targets for further downstream studies, including orthogonal validation. Consistent with previous results in bulk tissue, our results suggest a weak association between early-life exposure to Pb and DNA methylation, with some affected genes being related to neurodevelopment or cognitive function.

## Figures and Tables

**Figure 1 genes-10-00274-f001:**
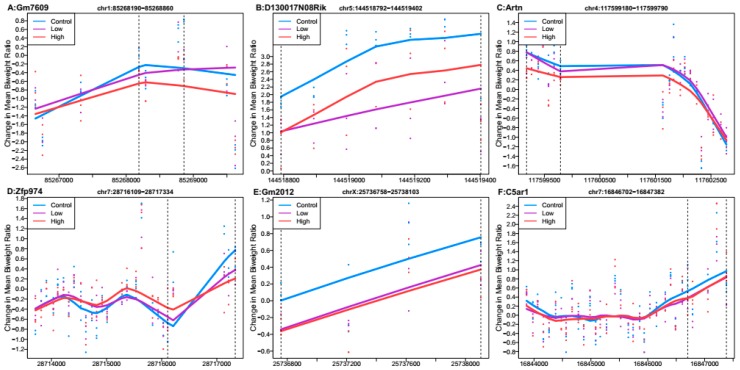
Differentially methylated regions associated with high perinatal Pb exposure (32 ppm) relative to controls. Top hits reaching Stouffer <0.5. Regions within dashed lines indicate the differentially methylated region. The area outside the boundary are plotted to show trends outside the region. Nearest gene to regions: (**A**) Predicted pseudogene 7609 (*Gm7609*), chromosome 1:85268190-85268860. (**B**) RIKEN cDNA D130017N08 gene (*D130017N08Rik*), chromosome 5:144518792-144519402. (**C**) Artemin (*Artn*), chromosome 4:117599180-117599790. (**D**) Zinc finger protein 974 (*Zfp974*), chromosome 7:28716109-28717334. (**E**) Predicted gene 2012 (*Gm2012*), chromosome X:25736758-25738103. (**F**) Complement component 5a receptor 1 (*C5ar1*), chromosome 7:16846702-16847382.

**Figure 2 genes-10-00274-f002:**
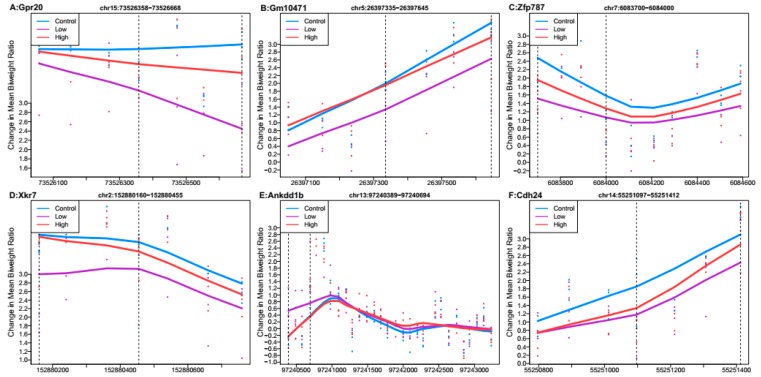
Differentially methylated regions associated with low perinatal Pb exposure (2.1 ppm) relative to controls. Regions within dashed lines indicate the differentially methylated region. Areas outside boundary are plotted to show the trend outside the region. Nearest gene to regions: (**A**) G protein-coupled receptor 20 (*Gpr20*), chromosome 15:73526358-73526668. (**B**) Predicted gene 10471 (*Gm10471*), chromosome 5:26397335-26397645. (**C**) Zinc finger protein 787 (*Xfp787*), chromosome 7:6083700-6084000. (**D**) X-linked Kx blood group related 7 (*Xkr7*), chromosome 2:152880160-152880455. (**E**) Ankyrin repeat and death domain containing 1B (*Ankdd1b*), chromosome 13:97240389-97240694. (**F**) Cadherin-like 24 (*Cdh24*), chromosome 14:55251097-55251412.

**Figure 3 genes-10-00274-f003:**
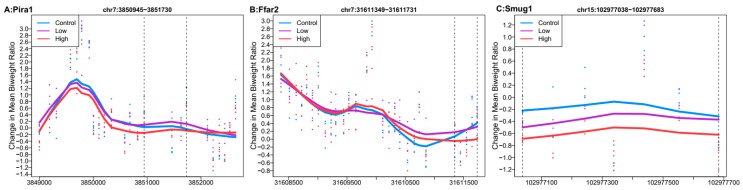
Differentially methylated regions overlapping between high (32 ppm) and low perinatal Pb exposure. Regions within dashed lines indicate the differentially methylated region. Areas outside the boundary are plotted to show the trend outside the region. Nearest gene to regions: (**A**) Paired-Ig-like receptor A1 (*Pira1*), chromosome 7: 3850945-3851730. (**B**) Single-strand selective monofunctional uracil DNA glycosylase (*Smug1*), chromosome 7:31611349-31611731. (**C**) Free fatty acid receptor 2 (*Ffar2*), chromosome 15:102977038-102977683.

**Table 1 genes-10-00274-t001:** Top gene ontologies enriched among differentially methylated genes associated with high perinatal Pb exposure (32 ppm) relative to control treated mice.

GO Term ID	Pathway	*p*-Value	Adjusted *p*-Value	Enrichment Score	Normalized Enrichment Score	Number of Permutations More Extreme	Size of Pathway
GO:0010165	response to X-ray	0.002	1	0.9	1.69	1	16
GO:0043206	fibril organization	0.0044	1	0.94	1.66	3	7
GO:0015693	magnesium ion transport	0.0052	1	0.91	1.68	4	11
GO:0014912	negative regulation of smooth muscle cell migration	0.0065	1	0.92	1.63	5	8
GO:0045197	establishment or maintenance of epithelial cell apical/basal polarity	0.0069	1	0.96	1.64	5	5
GO:0046887	positive regulation of hormone secretion	0.0071	1	0.87	1.64	6	14
GO:0051798	positive regulation of hair follicle development	0.0073	1	−0.69	−1.66	0	5
GO:0046545	development of primary female sexual characteristics	0.0073	1	−0.78	−1.88	0	5
GO:0045542	positive regulation of cholesterol biosynthetic process	0.0073	1	−0.68	−1.64	0	5
GO:0032471	reduction of endoplasmic reticulum calcium ion concentration	0.0073	1	−0.75	−1.81	0	5
GO:0050930	induction of positive chemotaxis	0.0074	1	0.91	1.65	6	10
GO:0010667	negative regulation of cardiac muscle cell apoptotic process	0.0074	1	0.91	1.63	6	9
GO:0071396	cellular response to lipid	0.0076	1	0.92	1.63	6	7
GO:0000186	activation of Mitogen-activated protein kinase kinase activity	0.009	1	0.84	1.6	8	23
GO:0001841	neural tube formation	0.0091	1	0.86	1.63	8	15

**Table 2 genes-10-00274-t002:** Top gene ontologies enriched among differentially methylated genes associated with low perinatal Pb exposure (2.1 ppm) relative to control treated mice.

GO Term ID	Pathway	*p*-Value	Adjusted *p*-Value	Enrichment Score	Normalized Enrichment Score	Number of Permutations More Extreme	Size of Pathway
GO:0007565	female pregnancy	0.001	0.97	0.58	1.51	0	56
GO:0006334	nucleosome assembly	0.001	0.97	0.59	1.56	0	85
GO:0042100	B cell proliferation	0.001	0.97	0.86	1.95	0	12
GO:0007218	neuropeptide signaling pathway	0.002	1	0.55	1.45	1	80
GO:0055003	cardiac myofibril assembly	0.0022	1	0.83	1.81	1	9
GO:0007379	segment specification	0.0024	1	0.91	1.75	1	5
GO:0055088	lipid homeostasis	0.0032	1	0.81	1.76	2	9
GO:0035567	non-canonical Wnt receptor signaling pathway	0.0032	1	0.83	1.8	2	9
GO:0006729	tetrahydrobiopterin biosynthetic process	0.0035	1	0.87	1.73	2	6
GO:0034695	response to prostaglandin E stimulus	0.0044	1	0.84	1.79	3	8
GO:0006105	succinate metabolic process	0.0044	1	0.84	1.78	3	8
GO:0044255	cellular lipid metabolic process	0.0045	1	0.86	1.78	3	7
GO:0060754	positive regulation of mast cell chemotaxis	0.0048	1	0.91	1.73	3	5
GO:0033600	negative regulation of mammary gland epithelial cell proliferation	0.006	1	0.9	1.71	4	5
GO:0019538	protein metabolic process	0.0075	1	0.77	1.69	6	10
GO:0002237	response to molecule of bacterial origin	0.0075	1	0.77	1.68	6	10
GO:0016051	carbohydrate biosynthetic process	0.0077	1	0.79	1.68	6	8
GO:0000186	activation of Mitogen-activated protein kinase kinase activity	0.008	1	0.66	1.62	7	23
GO:0042168	heme metabolic process	0.0094	1	0.85	1.69	7	6

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
