# Peer review of "Perinatal Lead (Pb) Exposure and Cortical Neuron-Specific DNA Methylation in Male Mice"

_genes, 2019, doi:10.3390/genes10040274_

Round 1
Reviewer 1 Report
This is a straight forward study exposing agouti mice to Pb at two relevant concentrations, through gestation and weaning, with offspring sacked at 10 months to assess neuronal methylation changes in the cortex of male mice. The strength of this paper lies in assessing Pb induced changes in methylation in a cell-type specific fashion. Many analyses do not reach genome-wide threshold and only males were used. I have a few questions for clarity before publication
Given the detailed discussion of the sex-specific nature of methylation profiles following Pb exposure, conducting this study in only males is a limitation. Presumably, both males and females were produced following gestational exposure, why were females not analyzed?
Were the offspring blood leads measured?
Beyond purity, what measures of neuronal viability were assessed following flow?
I would suggest additional references be considered. Studies that show hypermethylation following Pb exposure. Or studies relating to the fact that Pb has been shown to influence the function of zinc finger protein, it seems like an opportunity to discuss this. Also, in vitro, the effect of Pb on neuronal methylation extends beyond promotor regions. This seems worth noting including with references 53,54. https://doi.org/10.1093/toxsci/kfu028
For many analyses, no regions or pathways met genome-wide criteria. I appreciate your cautious interpretation of your data – “we interpret these suggestive results with caution.” Would you attribute this to the methodology focused on promotor methylation?
Author Response
Response to Reviewer 3 Comments
This is a straight forward study exposing agouti mice to Pb at two relevant concentrations, through gestation and weaning, with offspring sacked at 10 months to assess neuronal methylation changes in the cortex of male mice. The strength of this paper lies in assessing Pb induced changes in methylation in a cell-type specific fashion. Many analyses do not reach genome-wide threshold and only males were used. I have a few questions for clarity before publication
Given the detailed discussion of the sex-specific nature of methylation profiles following Pb exposure, conducting this study in only males is a limitation. Presumably, both males and females were produced following gestational exposure, why were females not analyzed?
We thank the reviewer for the opportunity to clarify on this issue. A part of the reason for discussing the possible sex-specific nature was to point out this limitation. We have more explicitly stated this is a limitation in the text saying, “a limitation of our study is it did not include female mice” and advise that “future studies should further explore the potential sex-specific effects of Pb”. Only males were analyzed to minimize sources of variation in a limited sample size in context of logistical and financial constraints.
Were the offspring blood leads measured?
Only maternal blood lead was measured. The 2.1 ppm (low) and 32 ppm (high) exposures corresponded to peak maternal blood levels of approximately 2.5 and 25 μg/dL, respectively. The exposure period was two weeks prior to mating, through gestation, and three weeks after birth. We declined to measure blood lead in offspring to avoid an additional stress exposure, particularly in the very young animals and we have added this statement to the methods section.
Beyond purity, what measures of neuronal viability were assessed following flow?
Brain tissue samples were frozen prior to processing and frozen cells have inherently damaged cell membranes. Cortex cells were further fixed and permeabilized to complete cell lysis prior to flow cytometry. We used FACS to sort individual nuclei, which were labeled with the neuron specific antibody (NeuN+). Our FACS side-scatter and forward-scatter dimensions were consistent with nuclei rather than cells. We did not flow sort intact cells and thus these organelles were non-viable prior to sort. We have clarified our methods section in that we conducted nuclei separation. We also found an error in the methods in the Percoll gradient step, and after confirming with lab notes, corrected the text to state we used the “bottom and middle layers” and not the top layer.
I would suggest additional references be considered. Studies that show hypermethylation following Pb exposure. Or studies relating to the fact that Pb has been shown to influence the function of zinc finger protein, it seems like an opportunity to discuss this. Also, in vitro, the effect of Pb on neuronal methylation extends beyond promotor regions. This seems worth noting including with references 53,54. https://doi.org/10.1093/toxsci/kfu028
Thank you for the suggestions for expanding references and discussion. For the paragraph discussion hypo/hypermethylation, we have expanded discussion to include additional studies:
“Early life Pb exposure in mice reduced cerebral levels of genes involved in maintenance of DNA methylation such as DNA Methyltransferase 1 (DNMT1) and Methyl-CpG Binding Protein 2 [53]. Pb exposure in a zebrafish model also inhibited DNMT1 activity and reduced global CpG methylation [54]. Other studies showed patterns of hypermethylation. In neocortex tissue of mice, exposure to Pb in utero was associated with more overall methylation and repression of gene expression [55]. Another study found the majority (>90%) of differentially methylated CpG sites in hippocampus of Pb exposed female mice were hypermethylated with less effect in males, and that effects of Pb were stronger in hippocampal tissue compared to cortex [56]. In contrast, in hippocampal tissue of rats the sex-linked Pb exposure observations were different, where 91% of differentially methylated genes were hypomethylated in females, but only 17% in males [57]. However, unlike the previous study conducted in hippocampal tissue, we observed a pattern of hypomethylation in cortex neuronal cells in a male only sample. Because our study did not include female mice, we are unable to examine sex differences. The predominance of hypomethylation we observed mirrors some previous research in Pb exposure and brain DNA methylation, but questions remain about potential cell type specific and sex-specific effects of Pb.”
We have also included discussion on Pb and zinc finger proteins:
“In both high and low Pb exposures, the top regions included a zinc finger protein. The zinc finger is a binding motif important in processes like DNA replication and transcription, and Pb competes for the zinc binding site [50]. Disruptions by Pb impact DNA binding of zinc finger proteins, and expression of target genes [51].”
We have also included the suggested reference for effects outside of promoter regions:
“Additionally, in vitro Pb exposure affects neuronal methylation in regions beyond promoters [50]. Promoters and CpG islands are potentially less responsive to the environment and more tightly regulated.”
For many analyses, no regions or pathways met genome-wide criteria. I appreciate your cautious interpretation of your data – “we interpret these suggestive results with caution.” Would you attribute this to the methodology focused on promotor methylation?
In addition to our limited sample size, we believe our use of a promotor tiling array method may be a contributing factor to lack of genome-wide significance. Areas outside of promoters and CpG islands have been observed to be highly responsive to environmental exposures. We advise future studies to consider this component in study design. “Future analyses may consider bisulfite sequencing with lower DNA input and greater genome coverage.”
Reviewer 2 Report
In this manuscript, Johu Dou and other authos used Roche NimbleGen Mouse DNA methylation 3x720K CpG island Promoter Array to detect methlyation difference between high/low Pb group and control in cortical neuronal nuclei of 10 month male mice cortical neuronal nuclei. The authors chose this tissue for it can reflect the long term effect in neuronal because of life span but the methlyation pattern in this tissue is subjected to neuronal activity, which cause some problem in decifiering the methylation difference. The manuscript is well writen and easy to be understood. And the study had a good design and the results are well explained. However, I still have some minor questions about it.
1. In page 2 Line 58-59, "A decrease in DNA methyltransferase activity (DNMT) was also observed in monkey brains", This sentence should be removed from here because I didn't find it connet to the context there. Besides, The introduction should be better organised to make it clear.
2. In page 2 Line 72, "These studies suggest that DNA methylation is a putative epigenetic mechanism by which early-life Pb exposure exerts lasting effects". Acctually, I think it's the tissue character of neuronal cells exert lasting effects because they have a long lifespan. Based on this character, we can see how Pb exposure caused lasting effects on neuronal cells.
3. In page 3 Line 128, I tried twice but I still can't open the Supplementary File downloaded from the MDPI website. I think the Supplementary file may be broken in the system.
4. In Figure 1 and Figure 3 . The author said the high Pb exposure (37ppm) but in page 2 Line 86 the high Pb exposure concentration is 32ppm.
Author Response
Response to Reviewer 1 Comments
In this manuscript, John Dou and other authors used Roche NimbleGen Mouse DNA methylation 3x720K CpG island Promoter Array to detect methylation difference between high/low Pb group and control in cortical neuronal nuclei of 10 month male mice cortical neuronal nuclei. The authors chose this tissue for it can reflect the long term effect in neuronal because of life span but the methylation pattern in this tissue is subjected to neuronal activity, which cause some problem in deciphering the methylation difference. The manuscript is well written and easy to be understood. And the study had a good design and the results are well explained. However, I still have some minor questions about it.
1. In page 2 Line 58-59, "A decrease in DNA methyltransferase activity (DNMT) was also observed in monkey brains", This sentence should be removed from here because I didn't find it connect to the context there. Besides, The introduction should be better organised to make it clear.
We have taken the reviewer’s suggestion and removed the specified line. We have moved that information to the discussion, as it is more relevant to a section’s content addressing later life changes: “Furthermore, in those aged monkeys a decrease in DNA methyltransferase activity (DNMT) was also observed [8], suggesting possible epigenetic changes through methylation.”
We have also slightly altered paragraph structure to make the transition into the topic of methylation clearer.
2. In page 2 Line 72, "These studies suggest that DNA methylation is a putative epigenetic mechanism by which early-life Pb exposure exerts lasting effects". Actually, I think it's the tissue character of neuronal cells exert lasting effects because they have a long lifespan. Based on this character, we can see how Pb exposure caused lasting effects on neuronal cells.
We have incorporated the insights of the reviewer and changed the beginning of the last paragraph of the introduction to read “Neuronal cells may be vehicles through which early-life Pb exposure exert lasting effects. Persistent epigenetic changes could be carried through the long lifespans of neurons.” Speaking about the long lifespans of neurons here also improves flow/better sets up the next lines about neuron specific analysis.
3. In page 3 Line 128, I tried twice but I still can't open the Supplementary File downloaded from the MDPI website. I think the Supplementary file may be broken in the system.
We apologize for the broken file. We have re-uploaded the zip file and checked that it could be opened after downloading from the system.
4. In Figure 1 and Figure 3 . The author said the high Pb exposure (37ppm) but in page 2 Line 86 the high Pb exposure concentration is 32ppm.
Thank you for catching this mistake, the actual concentration is 32ppm and the figure legends have been corrected.

Reviewer 3 Report
The authors investigate an important question about the molecular effects of early exposure to Pb on neurodevelopments. This study targets a very specific time window and exposure: during in utero and by mother’s drinking water, followed by assaying cortical neuron specific promoter methylation at adults. This reviewer has expertise in neurodevelopment and will primarily comment on this aspect of the research and its implication. The experiments are done in a well-controlled manner and inclusion of both low exposure and high exposure groups is appropriate and informative in the following analysis. Since the authors observed only weak associations between early Pb exposure (maternally) and DNA hypomethylation, it would be hard to be convinced about the significance of early exposure. At best, it would support the idea that early Pb exposure will put animals at risk for developing later cognitive or other brain malfunctions. This needs to be discussed more explicitly in the discussion part. Also, it would be helpful if the authors can discuss the literatures that compare human in utero Pb exposure and postnatal Pb exposure and how these different time windows affect later brain disorders. The reason is that embryonic development is very plastic and heavily mutated neurons are prone to apoptosis, so there is possibility that later, postnatal Pb exposure is more likely to cause more noticeable change of cognitive functions. It is interesting to raise this issue in the discussion to attract the attention of neurodevelopment biologists.
Some minor point: the authors should better organize the format of the manuscript. For example, in the Method section, the contents with Supplementary Figures should be moved to Results, especially the Supplementary Figure 1.
Author Response
Response to Reviewer 2 Comments
The authors investigate an important question about the molecular effects of early exposure to Pb on neurodevelopments. This study targets a very specific time window and exposure: during in utero and by mother’s drinking water, followed by assaying cortical neuron specific promoter methylation at adults. This reviewer has expertise in neurodevelopment and will primarily comment on this aspect of the research and its implication. The experiments are done in a well-controlled manner and inclusion of both low exposure and high exposure groups is appropriate and informative in the following analysis. Since the authors observed only weak associations between early Pb exposure (maternally) and DNA hypomethylation, it would be hard to be convinced about the significance of early exposure. At best, it would support the idea that early Pb exposure will put animals at risk for developing later cognitive or other brain malfunctions. This needs to be discussed more explicitly in the discussion part.
We have added a new paragraph into the discussion, where we point out the weak associations and discuss other avenues for Pb’s impacts.
“Overall, we observe only weak associations between early Pb exposure and DNA hypomethylation. At best, these Pb associated methylation changes may be involved in putting mice at risk for developing later cognitive abnormalities. Pb related impacts on the brain are not only through DNA methylation, but also through other impacts such as changes in brain structure. For example, in rats perinatal Pb exposure induced pathological changes in nerve endings [52]. Cumulative Pb exposure may also be an important consideration.”
We also moved the paragraph starting “While exposure to environmental toxicants may…” to be after this paragraph, as its content is related to the idea of later malfunctions.
Also, it would be helpful if the authors can discuss the literatures that compare human in utero Pb exposure and postnatal Pb exposure and how these different time windows affect later brain disorders. The reason is that embryonic development is very plastic and heavily mutated neurons are prone to apoptosis, so there is possibility that later, postnatal Pb exposure is more likely to cause more noticeable change of cognitive functions. It is interesting to raise this issue in the discussion to attract the attention of neurodevelopment biologists.
Continuing the previously mentioned paragraph, we also discuss literature on later life exposure and impacts.
“In human adults, cumulative occupational Pb exposure was associated with poorer performance in neuropsychological tests [53], and bone Pb concentration, reflecting cumulative exposure, is associated with faster declines in cognition test performance [4]. Pb exposure has clear brain impacts across the life course. In early life, Pb is associated with developmental delays, behavioral problems, and stunted growth, while in later life past Pb exposure is associated with cognitive decline [54].”
Some minor point: the authors should better organize the format of the manuscript. For example, in the Method section, the contents with Supplementary Figures should be moved to Results, especially the Supplementary Figure 1.
We have moved the content pertaining to the diagrams for FACS and sonication into the first section of the results. In text figures and tables have also been moved so that they are inserted into more appropriate locations. We appreciate all of the reviewer feedback.
